# Immune Checkpoint Inhibitors in Pre-Treated Gastric Cancer Patients: Results from a Literature-Based Meta-Analysis

**DOI:** 10.3390/ijms21020448

**Published:** 2020-01-10

**Authors:** Giandomenico Roviello, Silvia Paola Corona, Alberto D’Angelo, Pietro Rosellini, Stefania Nobili, Enrico Mini

**Affiliations:** 1Department of Health Sciences, University of Florence, 50139 Florence, ItalyEnrico.mini@unifi.it (E.M.); 2Department of Medical, Surgical and Health Sciences, University of Trieste, Cattinara Hospital-Strada di Fiume 447, 34149 Trieste, Italy; sil.corona@hotmail.it; 3Department of Biology and Biochemistry, University of Bath, Bath BA2 7AY, UK; ada43@bath.ac.uk; 4Unit of Medical Oncology, Department of Medicine, Surgery and Neurosciences, University of Siena, Viale Bracci-Policlinico “Le Scotte”, 53100 Siena, Italy; rosellini.pietro@gmail.com

**Keywords:** nivolumab, PD-1, PD-L1, immunotherapy, gastric cancer

## Abstract

Immunotherapy has recently changed the treatment of several cancers. We performed a literature-based meta-analysis of randomised controlled trials to assess the efficacy of the novel immune checkpoint inhibitors (ICIs) in metastatic gastric cancer. The main outcome was overall survival. Based on age (cut-off agreed at 65 years), tumour location (gastric vs. gastro-oesophageal junction), programmed death-ligand 1 (PD-L1) status, sex and Eastern Cooperative Oncology Group (ECOG) status (1 vs. 0), we scheduled a subgroup analysis for the overall survival. Three studies were included in the analysis for a total of 1456 cases (811 cases were in the experimental group and 645 cases in the control group). The pooled analysis showed improved overall survival in the experimental arm in the absence of statistical significance (hazard ratio (HR) = 0.87, 95% CI: 0.64–1.18; *p* = 0.37). The subgroup of patients with PD-L1-positive tumours (HR = 0.82 vs. 1.04) and gastro-oesophageal junction cancer (HR = 0.82 vs. 1.04) showed a statistically significant advantage of overall survival. This study supports the efficacy of immune checkpoint inhibitors in the subgroup of patients with metastatic gastric cancer with PD-L1-positive and gastro-oesophageal junction tumour location. Future studies are needed with the aim of identifying reliable predictive biomarkers of ICI efficacy.

## 1. Introduction

Gastric cancer (GC) accounted for over 1,000,000 new cases in 2018 with an estimated 783,000 deaths, making it the fifth most common cancer worldwide and the third leading cause of cancer deaths [1]. Metastatic GC is managed with a combination of chemotherapy composed of platinum compounds plus fluoropyrimidines [2,3]. For patients with progression of disease on first-line chemotherapy, treatment options include chemotherapy with irinotecan, taxanes (paclitaxel or docetaxel) and ramucirumab (a monoclonal antibody against vascular endothelial growth factor 2 (VEGFR2) used in monotherapy or in combination with paclitaxel) [4,5]. Although no standard treatment is recommended by guidelines for patients who failed two or more lines of therapy, two drugs, apatinib (a tyrosine kinase inhibitor that selectively inhibits VEGFR2) and trifluridine/tipiracil (TAS-102), were shown to increase survival when compared to a placebo as third-line or later therapy for advanced GC [6,7]. Nonetheless, the prognosis of these patients remains very poor, underlying the need to develop new treatment options with acceptable safety profiles.

Immunotherapy with immune checkpoint inhibitors (ICIs) has revolutionized the treatment of several solid and haematological cancers. ICIs work by inhibiting programmed death-1 (PD-1) receptor or ligand (PD-L1) [8]. There is a strong rationale for the use of ICIs for advanced GC; in fact, up to 65% of gastric cancers overexpress PD-L1 [9,10,11]. Based on the results of a phase II study [12], pembrolizumab (an anti-PD-1 ICI) has been approved by the food and drug administration (FDA) for the treatment of patients with recurrent locally advanced or metastatic GC whose tumours express PD-L1. However, in the following phase III trial, no remarkable overall survival (OS) outcomes have been reported with pembrolizumab when compared with paclitaxel as a second-line regimen in PD-L1-positive advanced gastric cancer or gastro-oesophageal junction cancer [13]. Additional phase III studies on ICIs for metastatic GC showed discordant results [14,15] and nivolumab has been approved in Japan [16]. In this paper, we performed a meta-analysis to evaluate the impact of ICIs on the outcomes of patients with metastatic GC focusing on the possible predictor of efficacy.

## 2. Results

### 2.1. Literature Review and Characteristics of the Included Studies

The search yielded 770 potentially relevant articles. A total of 506 studies were excluded as duplicates. After viewing the titles and abstracts of the 264 remaining studies, the full texts of 13 studies were retrieved, and three studies [13,14,15] were included in the analysis (Figure 1). Table 1 summarized the characteristics of the evaluated studies. A total of 1456 cases were included: the experimental arm included 811 cases whereas the control arm included 645 cases. Two studies [13,14] investigated a PD-1 ICI as an experimental drug, while the placebo was the control arm of one study [14]. All studies were phase III randomised controlled trials (RCTs). The mean Jadad score was 4.3 (Table 1). No publication bias was estimated on account of the small number of included trials.

### 2.2. Efficacy Data

Table 2 and Table 3 summarize characteristics of patients and efficacy data of the evaluated studies. The pooled analysis showed improved OS in the experimental arm, in the absence of statistical significance (hazard ratio (HR) = 0.87, 95% CI: 0.64–1.18; *p* = 0.37, Appendix A). A random-effects model was used to perform the analysis (I^2^ = 87%). Furthermore, the pooled analysis showed no improvement in progression free survival (PFS) with the use of ICIs, in comparison to no ICIs (HR = 1.16, 95% CI: 0.62–2.17; *p* = 0.65, Appendix A). A random-effects model was used to perform the analysis (I^2^ = 97%). Lastly, each study was investigated for the response rate: 67 out of 749 (8.9%) patients for the experimental arm and 45 out of 613 (7.3%) patients for the control arm showed a tumour response. Using the Mantel–Haenszel method, the pooled response rate was 1.38 (95% CI 0.30–6.46; *p* = 0.68; I^2^ = 79%; Figure 2). The analysis was performed using a random-effects model and did not reach any statistical significance. Data on disease control rate could be obtained only from two studies [14,15], and showed disease control in a total of 142/453 (31.3%) patients in the experimental arm, and in a total of 103/317 (32.4%) patients in the control group. For disease control rate, a meta-analysis was not performed.

According to the trial protocol, a subgroup analysis was performed to investigate the influence of PD-L1 status, age, tumour location, sex and Eastern Cooperative Oncology Group (ECOG) performance status on OS (Table 4). The analysis of OS according to PD-L1 tumour expression status showed that ICIs significantly improved OS in patients with PD-L1-positive tumours (HR = 0.82 95% CI: 0.67–0.99; *p* = 0.04) in comparison to PD-L1-negative patients (HR = 1.04 95% CI: 0.77–1.42; *p* = 0.80) Figure 2. In addition, when we stratified patients according to the tumour location, we found that OS was significantly higher in the experimental arm versus the control arm in patients with gastro-oesophageal junction tumours (HR = 0.67 95% CI: 0.48–0.92; *p* = 0.01) compared to gastric tumours (HR = 0.92 95% CI: 0.66–1.29; *p* = 0.63) Figure 3. Lastly, there were no other differences when applying other stratification factors (Table 4).

In regard to the toxicity, a total of 386/808 (47.8%) patients for the ICIs and a total of 406/614 (66.1%) patients in the control arm reported all-grade treatment-related adverse events (Appendix A). Using the Mantel–Haenszel method for combining trials, the pooled risk ratio was 0.86 (95% CI 0.54–1.36; *p* = 0.51; I^2^ = 95%; Appendix A). The analysis was performed using a random-effects model. In addition, a total of 94/808 (11.6%) patients for the ICIs and a total of 172/614 (28%) patients in the control arm reported all-grade treatment-related adverse events (Appendix A). A pooled analysis showed a risk ratio of 0.59 (95% CI 0.22–1.55; *p* = 0.28; I^2^ = 91%; random-effects model Appendix A). These data showed a non-statistically significant trend in favour of a lesser toxicity for ICIs.

## 3. Discussion

The treatment of several tumours has been revolutionized by the introduction of ICIs [17], and for this reason, they have been evaluated in metastatic GC. Unfortunately, the here-presented meta-analysis, including three RCTs with a total of 1456 pre-treated metastatic GC patients, and targeted with ICIs, suggests that there is no statistically significant improvement in OS, and no advantage in terms of PFS and tumour response rate. However, since this meta-analysis analysed only three RCTs, it is not possible to conclude that ICIs do not work in pre-treated metastatic GC. In addition, it should be considered that endpoints, such as PFS and response rate (RR), are not optimal in patients treated with ICIs [18].

A high heterogeneity (I^2^ = 87%) was reported between the studies, which could be explained by differences in the patient population (Table 1 and Table 3). In fact, it is well-known that patient ethnicity affects prognosis in GC patients [19]. The ATTRACTION-2 trial enrolled only Asian patients (mainly from Japan), while the JAVELIN Gastric 300 and KEYNOTE-61 trials enrolled patients coming from Europe, and in a smaller percentage from Asian countries. In addition, not all studies stratified patients according to PD-L1 expression. Lastly, there were differences in the definition of the control arm, for which a placebo was used in one study and chemotherapy in the other two.

Nonetheless, the negative results in terms of OS observed in our meta-analysis highlight the pressing need for stringent patient recruitment and the identification of those who might benefit most from ICI treatment. Recently, the Cancer Genome Atlas categorised GCs into four molecular subtypes: Microsatellite instability (MSI)-high, Epstein–Barr virus-positive, genomically stable, and chromosomally unstable [9]. If we use this molecular classification, MSI-high and Epstein–Barr virus-positive are the tumour subtypes more likely to respond to ICIs: MSI-high, in view of the increased number of somatic mutations due to DNA mismatch repair defects, and Epstein–Barr virus-positive for the alterations related to the presence of oncogenic viruses [20]. Interestingly, two subgroups of patients showed a statistically significant OS advantage with ICIs: PD-L1 tumour-positive patients when compared to PD-L1 tumour-negative patients (HR: 0.82 vs. 1.04 respectively, Figure 2), and patients with gastro-oesophageal junction tumours when compared to gastric tumours (HR: 0.67 vs. 0.92 respectively, Figure 3). The low heterogeneity (0% for the PD-L1-positive subgroup and 23% for gastro-oesophageal junction location) further supports this data.

PD-L1 overexpression is generally associated with worse survival in patients with solid tumours [21]. However, despite the efficacy of immunotherapy in several solid and non-solid cancers, it is not well defined whether PD-L1 expression correlates also with the clinical response and outcomes in patients treated with ICIs. In 2016, a meta-analysis of 20 trials involving patients with melanoma, non-small cell lung cancer and renal cell carcinoma receiving anti-PD-1/PD-L1 antibodies showed that PD-L1 expression is significantly associated with risk of death and clinical response to anti-PD-1/PD-L1 ICIs in melanoma patients. Also, PD-L1 expression is associated with clinical response in patients with non-squamous non-small cell lung cancer [22]. PD-L1 positivity is observed in up to 65% of GC specimens [20], and even though it is widely associated with negative features, such as tumour size, lymph node metastasis and depth of invasion, there is no consensus on the role of PD-L1 expression as a prognostic biomarker in GC [23]. While the KEYNOTE-061 trial confirmed a better outcome with pembrolizumab in patients with PD-L1-positive tumours [13] data from the JAVELIN Gastric 300 and ATTRACTION-2 trials did not support the concept of PD-L1 positivity as a predictive response marker to ICIs [14,15]. In addition, PD-L1 expression was prospectively assessed in the KEYNOTE-061 and JAVELIN Gastric 300 trials, but was retrospectively evaluated in the ATTRACTION-2 trial. In this context, our meta-analysis seems to confirm a survival benefit for patients with PD-L1-positive GCs (Figure 2). However, it should be noted that the difference in methods (distinct immunohistochemistry antibody clones, staining methods and scoring systems) and the difference in PDL1 expression scoring systems (Table 1) do not allow for the drawing of definitive conclusions on the role of PD-L1 expression as a biomarker for the identification of patients who are likely to benefit from ICIs.

PD-L1 expression was measured in tumour cells in the ATTRACTION-2 and JAVELIN Gastric 300 trials, while PD-L1 expression was measured in tumour cells, lymphocytes and macrophages in the KEYNOTE-061 trial. Therefore, intra-tumour heterogeneity may influence the results of PD-L1 expression [24]. However, supplementary data from the JAVELIN Gastric 300 trial reported comparable results when PD-L1 expression was measured in both tumour and immune cells [15] (data not shown). For this reason, more prospective studies are needed.

We also found a statistically significant advantage for OS in patients with gastro-oesophageal junction tumours when compared to gastric tumours (HR: 0.67 vs. 0.92 respectively; Figure 3). Although the site of the primary GC in the upper third of the stomach, particularly at the gastro-oesophageal junction or cardia, is generally considered to be a prognostic factor [25], few data are available on the role of ICIs in unselected patients with gastro-oesophageal junction cancers. Interestingly, a very low percentage of gastro-oesophageal junction cancer patients account for the MSI-high and Epstein–Barr virus-positive molecular subtypes [9], implying that other clinical/molecular features may be responsible for the higher sensitivity to ICIs, which appears to characterise gastro-oesophageal junction tumours. Finally, some concerns may derive from the fact that we analysed two trials with an anti-PD-1 ICI and one trial with an anti-PD-L1 ICI, and it should be supposed that one is better than the other. In this context, a recent meta-analysis in pre-treated non-small-cell lung cancer patients indicated a slight benefit from anti-PD-1 than from anti-PD-L1 inhibitors [26]; however, in the absence of a direct comparison between anti-PD-1 and anti-PD-L1, no definitive conclusions are available, and we should consider all groups of ICIs to be of equal efficacy.

There are limitations to our study: (1) the analysis was based on the literature rather than on individual patient data; (2) a small number of studies were available; (3) heterogeneous disease characteristics, therapy regimens and heterogeneity of disease features might have an impact on outcome estimates; (4) an adverse events analysis was not done; (5) discrepancies within control arms among the studies (placebo or chemotherapy) have been reported; and (6) the absence of data on metastatic esophageal cancer.

To date, even with the availability of novel therapies [27], the prognosis of metastatic GC remains very poor. Immunotherapy has impressively modified the treatment strategy of several cancers and shown some efficacy in metastatic GC, especially in patients with PD-L1-positivity and gastro-oesophageal junction tumour location. We strongly suggest that subsequent clinical trials focus on the identification of predictive response biomarkers for a better selection of the optimal candidates for ICIs therapies.

## 4. Methodology

### 4.1. Data Retrieval Strategies

We analysed RCTs according to the Preferred Reporting Items for Systematic Reviews Guidelines and Meta-Analyses (PRISMA). The Web of Sciences (WOS), PubMed and Embase databases were analysed for suitable publications with the search strategy here reported (Appendix A). The search criteria were limited to phase III RCTs and studies published before December 2018 were considered.

### 4.2. Inclusion Criteria

Selected trials were independently examined by two authors (P.R., A.D.G.), according to the specific selection and exclusion criteria. Contentious points were resolved by referring to the corresponding authors. The main inclusion criteria were: (1) patients with advanced/metastatic GC that progressed after one line of therapy for metastatic disease; (2) a novel immune checkpoint inhibitor as the experimental arm; (3) a placebo or chemotherapy as a control arm for comparison; (4) OS as primary outcome and PFS as secondary outcome both showed as HR and RR (partial plus complete response) and disease control rate (DCR) (partial plus complete response plus stable disease) both showed as the number of patients out of the total who reported a documented tumour response. The following exclusion criteria were used: (1) non-randomised clinical trials; (2) <50 patients for an arm; (3) insufficient data available to estimate the outcomes; (4) studies in the first line of treatment for metastatic GC; and (5) studies that involved ICIs in both the experimental and control arms.

### 4.3. Data Collection and Quality Assessment

Hazard ratios for OS and PFS, the number of patients who reported a disease response or disease control and characteristics of analysed studies were independently extracted by two authors. The ICI was considered the experimental arm. The Jadad 5-item scale was used for study quality [28]. The protocol of the systematic review here employed is fully available at http://www.crd.york.ac.uk/PROSPERO, and it has been registered on the PROSPERO: International prospective register of systematic reviews (accession number CRD42019120737).

### 4.4. Statistical Analysis

RevMan 5.3 software was used for statistical analysis. Based on the absence or presence of heterogeneity, a random-effects model (the DerSimonian–Laird method) or a fixed-effects model (the Mantel–Haenszel method) [29,30] was used to generate the summary estimates. The I^2^ statistic, with percentage values of 25%, 50% and 75%, showing low, moderate and high heterogeneity, respectively, and a Q-test, were used to assess statistical heterogeneity [31]. The fixed-effects model, or alternatively, the random-effects model, was employed when *p* > 0.1 and I^2^ < 50% were achieved. For each study, time-to-event variables, including HRs with 95% CI, OS, PFS and dichotomous variables, such as RRs with 95% CIs, were calculated. Statistical significance was agreed at the value of *p* < 0.05. According to PD-L1 status, age (65 years was the cut-off), tumour location (gastric vs. gastro-oesophageal junction), sex and ECOG performance status (1 vs. 0), we scheduled a subgroup analysis for OS.

## Figures and Tables

**Figure 1 ijms-21-00448-f001:**
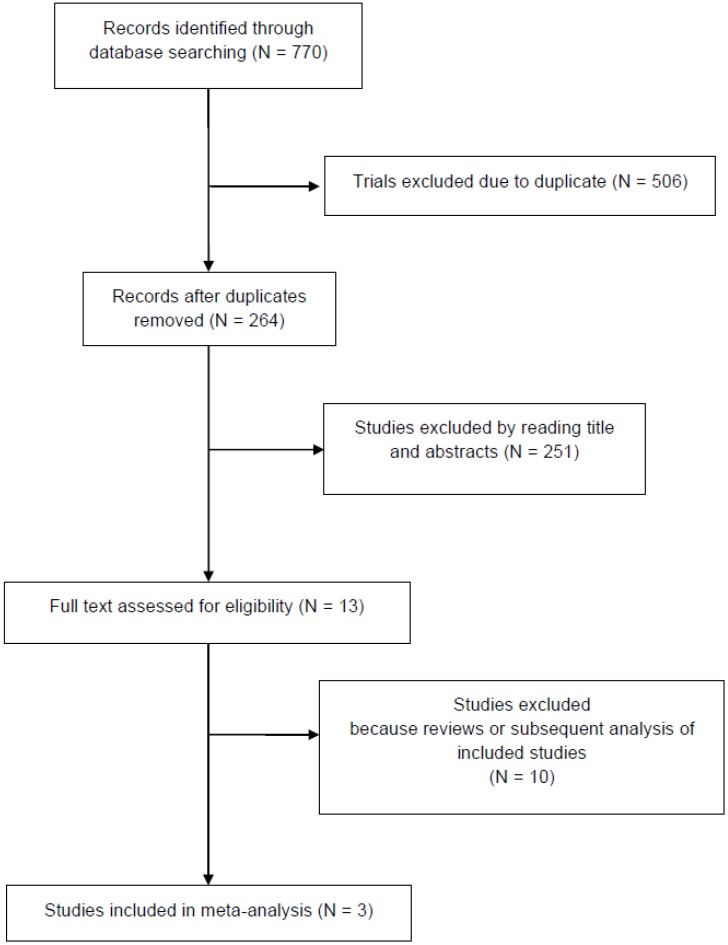
The trial selection flow chart.

**Figure 2 ijms-21-00448-f002:**
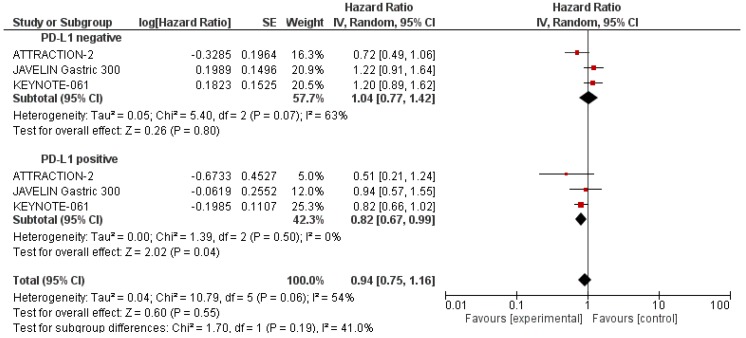
Subgroup analysis for overall survival of immune checkpoint inhibitors compared to the control arm in PD-L1-positive and PD-L1-negative patients.

**Figure 3 ijms-21-00448-f003:**
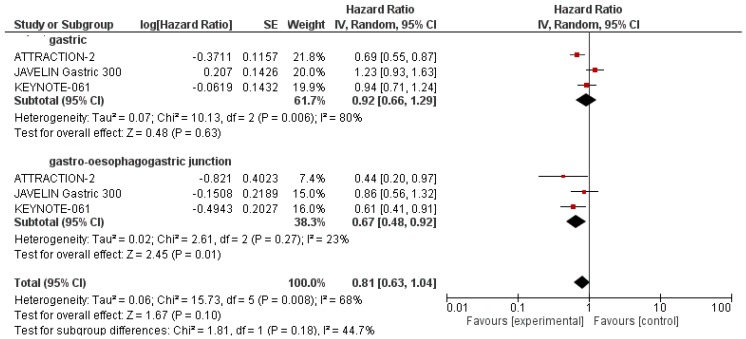
Subgroup analysis for overall survival of immune checkpoint inhibitors compared to the control arm in patients with gastro-oesophagogastric junction and gastric tumour location.

**Table 1 ijms-21-00448-t001:** Characteristics of the analysed trials.

Study	Primary Endpoint	Number of Patients Experimental Arm	Number of Patients Control Arm	Experimental Drug/Control Arm	Geographic Region	Line of Therapy	Programmed Death-Ligand 1 (PD-L1) Positivity	Jadad Score
ATTRACTION-2	OS	330	163	Nivolumab/Placebo	Asian	III/IV/V	staining in 1% or more of tumour cells	5
JAVELIN Gastric 300	OS	185	186	Avelumab/Chemotherapy	European/North America/Asian	III	by using an immunohistochemistry-based companion diagnostic (PD-L1 pharmDx) test	4
KEYNOTE-061	OS/PFS	296	296	Pembrolizumab/Chemotherapy	European/North America/Asian	II	PD-L1 combined positive score CPS of 1 or higher	4

OS: overall survival, PFS: progression free survival.

**Table 2 ijms-21-00448-t002:** Data on overall survival, progression-free survival and tumour response of the included studies.

Study	OS (Months)		PFS (Months)		Overall Response Rate (%)		Disease Control Rate (%)		Treatment Duration of Experimental Drug (Months)
	*Exp Arm*	*C Arm*	*Exp Arm*	*C Arm*	*Exp Arm*	*C Arm*	*Exp Arm*	*C Arm*	*Exp Arm*
ATTRACTION-2 2017	5.26	4.14	1.61	1.45	11.2	0	40.3	25	1.92
JAVELIN Gastric 300	4.6	5.5	1.4	2.7	2.2	4.3	22.2	44.1	8.0 *
KEYNOTE-061	6.7	8.3	1.5	4.1	11.1	12.5	NR	NR	4.4

Exp: experimental; C: control; NR: not reported *: weeks.

**Table 3 ijms-21-00448-t003:** Characteristics of patients in the evaluated studies.

Study	Median Age/Male Patients %		ECOG > 0%		Diffuse Histolog %		Primary Lesion GEJ Junction %		Prior Surgery %		Number of MetastaticSites > 2 %		Peritoneal Metastasis %		>II Previous Line of Treatment %		Previous Ram. %	
	*E*	*C*	*E*	*C*	*E*	*C*	*E*	*C*	*E*	*C*	*E*	*C*	*E*	*C*	*E*	*C*	*E*	*C*
ATTRACTION-2	62/69	61/73	71	71	NR	NR	NR	NR	60 °	64 °	75 °°	73 °°	19	26	79	72	11	13
JAVELIN Gastric 300	59/76	61/68	64	67	NR	NR	34	26	NR	NR	NR	NR	NR	NR	0	0	NR	NR
KEYNOTE-061	62/68	60/70	57	53	29	22	30	32	36	38	NR	NR	28	28	0	0	NR	NR

E: experimental arm; C: CONTROL arm; ECOG: Eastern Cooperative Oncology Group; NR: not reported; GEJ: gastro-oesophageal; Ram: ramucirumab; ° Gastrectomy; °° ≥ 2 organs with metastases.

**Table 4 ijms-21-00448-t004:** Subgroup analysis of immune checkpoint inhibitors (ICIs) compared to the control arm in gastric cancer.

	HR	95% CI	*p* Value	I^2^ %	*p* Value	Model
PD-L1 positive PD-L1 negative	0.82 1.04	0.67–0.99 0.77–1.42	0.04 * 0.80	0 63	0.50 0.07	Random
Age ≥ 65 years Age < 65 years	0.76 0.89	0.52–1.11 0.66–1.19	0.16 0.41	70 67	0.04 0.05	Random
Male Female	0.79 1,01	0.58–1.08 0.67–1.51	0.14 0.98	76 63	0.02 0.07	Random
ECOG:0 ECOG:1	0.84 0.87	0.50–1.41 0.67–1.13	0.51 0.29	82 63	0.004 0.07	Random
Gastric Gastro-oesophageal Junction	0.92 0.67	0.66–1.29 0.48–0.92	0.63 0.01 *	80 23	0.006 0.27	Random

HR: hazard ratio; * Statistically significative

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
