# Peer review of "Immune Checkpoint Inhibitors in Pre-Treated Gastric Cancer Patients: Results from a Literature-Based Meta-Analysis"

_ijms, 2020, doi:10.3390/ijms21020448_

Round 1
Reviewer 1 Report
This is a fairly well written article based on a meta-analysis of randomized controlled trials for evaluating the efficacy of the novel ICIs in metastatic gastric cancer patients. I have only some minor comments:
pag. 1, line 33: authors should include updated epidemiological data on gastric cancer, referring to GLOBOCAN 2018 report; pag.1, line 34:it should be “combination chemotherapy” instead of “combination of chemotherapy”; pag. 1, line 40: authors should remove the link to Wikipedia unless it has been intentionally introduced in the text; pag. 2 line 23: as mentioned in the text, two authors, i.e. PR and ADG independently examined selected trials. Authors should include their contribution in the specific section (pag.8); pag. 4, fig.1: authors should specify reasons for exclusion of studies (N=10), as for the other studies and trials indicated in the flow chart; pag 8, line 29: it should be “trial” instead of “trail”.
Author Response
pag. 1, line 33: authors should include updated epidemiological data on gastric cancer, referring to GLOBOCAN 2018 report;
It has been updated
pag.1, line 34:it should be “combination chemotherapy” instead of “combination of chemotherapy”;
It has been changed
pag. 1, line 40: authors should remove the link to Wikipedia unless it has been intentionally introduced in the text;
It has been removed
pag. 2 line 23: as mentioned in the text, two authors, i.e. PR and ADG independently examined selected trials. Authors should include their contribution in the specific section (pag.8);
It has been added
pag. 4, fig.1: authors should specify reasons for exclusion of studies (N=10), as for the other studies and trials indicated in the flow chart;
Reasons have been included
pag 8, line 29: it should be “trial” instead of “trail”
It has been changed
Reviewer 2 Report
In the article “immune checkpoint inhibitors and pretreated gastric cancer patients-results from the literature based meta-analysis” the authors try to evaluate the role of checkpoint inhibitors in the pretreated metastatic gastric and gastroesophageal cancers to assess the efficacy in terms of survival. Following are my comments.
1. What is the rationale for not including metastatic esophageal cancer as they are also treated on the similar lines in the metastatic setting?
2. Checkmate 032 is a randomized study for advanced gastric esophageal or esophageal gastric cancer assessing the role of nivolumab. Any reason for not including this study? What about keynote=0181?
3. The whole manuscript did not see anything about the safety data, without which it will be considered as incomplete.
4. In the abstract correct the typos with respect to ECOG status. He is at 1 versus 0?
5. In the abstract what is the light actually indicate with respect to the subgroup of patients with PDL 1+ tumors? The line just ended saying status tickly significant advantage. Statistically significant advantage of what?
6. Following this line, and abstract, it was noted that the study supported the efficacy of checkpoint inhibitors in the subgroup of patients of PDL 1+ and gastroesophageal junction metastatic gastric cancer. What is this is gastric esophageal junction metastatic gastric cancer?
7. Under the subsection 3.2 efficacy data the authors noted improvement survival in experimental arm in the absence of status to the significance. It would be more clear and a purely if actual numbers of survival could be mentioned rather than just leaving it up as status to kill significance or not.
8. The foot note for the figure 3 mention gastric for esophageal junction positive and gastric location. This needs to be further clarified.
9. Page #8, third paragraph. Authors mention that the found status needs significant advantage for patients with gastroesophageal junction tumors when compared to gastric tumors. Statistician significant advantage for water park?
10. The last paragraph-The font was inappropriately large.
It appears overall the manuscript needs substantial revision in order to get this to the level of publication.
Author Response
What is the rationale for not including metastatic esophageal cancer as they are also treated on the similar lines in the metastatic setting?
We decided to evaluate the role of ICIs only in GC because, as reported in several phase III studies, gastric and esophageal cancer are treated as two different entities. Generally, only patients with gastroesophageal junction carcinoma are included in trials for GC. We are planning another meta-analysis for esophageal cancer, and we reported this caveat in limitations of the study section.
Checkmate 032 is a randomized study for advanced gastric esophageal or esophageal gastric cancer assessing the role of nivolumab. Any reason for not including this study? What about keynote=0181?
According to inclusion criteria, this meta-analysis evaluated chemotherapy or placebo but not immunotherapy as control arm whereas Checkmate 032 trial had nivolumab in two different schedules as control arm. The keynote 181 has been excluded because enrolled only advanced esophageal cancer. The methodology has been updated.
The whole manuscript did not see anything about the safety data, without which it will be considered as incomplete.
According to your suggestion we added the data of any grade and grade ≥3 treatment-related adverse events.
In the abstract correct the typos with respect to ECOG status. He is at 1 versus 0?
It has been changed
In the abstract what is the light actually indicate with respect to the subgroup of patients with PDL 1+ tumors? The line just ended saying status tickly significant advantage. Statistically significant advantage of what?
The sentence refers to OS, abstract has been changed accordingly.
Following this line, and abstract, it was noted that the study supported the efficacy of checkpoint inhibitors in the subgroup of patients of PDL 1+ and gastroesophageal junction metastatic gastric cancer. What is this is gastric esophageal junction metastatic gastric cancer?
The sentence refers to the subgroup of patients with metastatic gastric cancer gastroesophageal junction tumor location, the sentence has been changed accordingly.
Under the subsection 3.2 efficacy data the authors noted improvement survival in experimental arm in the absence of status to the significance. It would be more clear and a purely if actual numbers of survival could be mentioned rather than just leaving it up as status to kill significance or not.
In this sentence we reported the HR that expressed the meaningful advantage of using ICIs compared to control arm. In addition, we reported the data of median OS for all included trials.
The foot note for the figure 3 mention gastric for esophageal junction positive and gastric location. This needs to be further clarified.
The word “positive” is a typo and has been deleted.
Page #8, third paragraph. Authors mention that the found status needs significant advantage for patients with gastroesophageal junction tumors when compared to gastric tumors. Statistician significant advantage for water park?
We referred to OS, the sentence has been changed accordingly.
The last paragraph-The font was inappropriately large.
It has been changed
Reviewer 3 Report
Dear authors,
Author Response
Thank you for your comments
Round 2
Reviewer 2 Report
thank you for addressing the queries.